# Whole Blood Transcriptome Analysis Reveals Positive Effects of Dried Olive Pomace-Supplemented Diet on Inflammation and Cholesterol in Laying Hens

**DOI:** 10.3390/ani9070427

**Published:** 2019-07-07

**Authors:** Marco Iannaccone, Andrea Ianni, Solange Ramazzotti, Lisa Grotta, Elettra Marone, Angelo Cichelli, Giuseppe Martino

**Affiliations:** 1Faculty of Bioscience and Technology for Food, Agriculture, and Environment, University of Teramo, Via R. Balzarini 1, 64100 Teramo, Italy; 2Department of Medical, Oral and Biotechnological Sciences, G. D.’Annunzio University, Via dei Vestini,31, 66100 Chieti, Italy

**Keywords:** dried olive pomace, RNA-sequencing, laying hens

## Abstract

**Simple Summary:**

Olive pomace (OP) represents an important source of bioactive compounds which have been successfully used for animal nutrition. In this study, we elucidate the whole transcriptome of laying hens fed with a dried OP (DOP)-supplemented diet using an RNA sequencing approach. We found that DOP modulates several biological pathways mainly related to inflammatory response and cholesterol biosynthesis. Consistent with the gene expression data, we noted a decrease of egg yolk cholesterol. Thus, our study provides evidence that a DOP-supplemented diet improves egg quality and, at the same time, ameliorates inflammatory animal status.

**Abstract:**

Olive pomace (OP) represents one of the by-products of the olive industry and represents an important source of bioactive compounds. This characteristic makes OP a potential feed supplement in livestock nutrition. Thus, in the last years, several studies have been published to evaluate the productive traits following OP supplementation in animal diets; however, relatively little is known from a molecular biology standpoint. Therefore, in this study, we report the RNA-sequencing analysis of laying hens fed with a 10% dried OP (DOP) supplementation. Applying a false discovery rate (FDR) <0.05 and a Log_2_Fc either less than −1.5 or higher than +1.5, we identified 264 differentially regulated genes (DEGs) between the non-supplemented diet control group (CTR) and the DOP group. Using the 264 DEGs to identify enriched biological pathways, we noted that cholesterol biosynthesis showed the highest enrichment followed by several pathways related to immune response and inflammation. As a consequence, when we quantified the cholesterol amount in yolk egg, we found a significant reduction in the DOP vs. the CTR group (*p* < 0.05). In conclusion, this study shows that DPO affects gene expression in laying hens, which is directly correlated with cholesterol decrease and can potentially ameliorate health status influencing immune response and inflammation.

## 1. Introduction

The cultivation of olive trees constitutes one of the main economic resources for agriculture, as olive trees are cultivated on a surface of about 10 million hectares in more than 40 countries [1]. The European community, and in particular the countries located in the Mediterranean area, contributes about 70% of the global olive production (https://ec.europa.eu/agriculture/olive-oil/economic-analysis_en.pdf). Olive fruits are composed of about 50% water while the remainder is equally divided into oil and dry matter [2]; olives are either consumed as table olives or they are industrially processed for oil production. Both usages produce a huge amount of solid wastes and dark liquid effluents. Between the different wastes generated by the olive industry, olive pomace (OP) is the main residue of the olive oil extraction process. It is the pulpy material remaining after removing most of the oil from the olive paste and it consists of pieces of pulp (70–90%), stone (9–27%), and olive kernel (2–3%) [3]. OP is attracting more and more interest because—to a large extent—it contains valuable substances such as carbohydrates, organic acids, mineral nutrients, oils, fibers, and phenols that can be used for different purposes such as supplementation for both human and animal diets [4]. Indeed, agro-industrial by-products like grape pomace have been used successfully as diet supplements to improve the quality and oxidative status of milk [5] as well as to modify the aromatic profile of fresh cheese [6] in Fresian cows. Also in dairy cows, OP at 10% was used as a diet supplement and, while there was no difference in milk yield or composition (except protein), OP was found to positively affect milk nutritional and nutraceutical properties, reducing atherogenic and thrombogenic indices in addition to increasing conjugated linoleic acid percentage [2]. Furthermore, different studies have evaluated the possibility of including OP in the diet of ruminants. This was found to be a favorable practice because the neither body weight nor milk yield were negatively influenced [7,8]. Moreover, a positive correlation between OP supplementation and ameliorated product quality, mostly in terms of fatty acid (FA) composition and percentage of active secondary metabolites, was also shown [9].

In different species like chicken, the use of organic and/or natural additives with antioxidant properties is a common practice because the richness of polyunsaturated fatty acids makes the meat particularly prone to oxidation. For example, Branciari et al. [10] demonstrated that adding polyphenol-rich olive cake to the animals’ diet ameliorated both the oxidative stability of the chicken meat and the quality of the meat. When polyphenols were supplemented in drinking water, positive effects on oxidative status were also recorded [11]. Indeed, the expression of commonly used oxidative stress markers such as thiobarbituric acid reactive species (TBARS), superoxide dismutase activity (SOD), and glutathione (GSH) were all reduced. In laying hens, Cayan and Erener [12] recorded a decreased tendency in eggs cholesterol amount when olive leaf powder was added to the animals’ diet and the productive performance of the hens was measured.

All these studies encourage about the use of olive by-products as a feed supplement in poultry farms. However, none of them have investigated the effects of the studied compounds on the host’s transcriptome. Therefore, together with the evaluation of animal performances, it is important also to assess the molecular mechanism and the metabolic pathways behind the productive traits for the pruposes of fine-tuning the nutritional management of livestock. Thus, in this study, driven by our hypothesis that an OP-supplemented diet could affect gene expression and consequently metabolism-related pathways, we profiled the whole blood transcriptome of laying hens using an RNA-sequencing (RNA-seq) approach.

## 2. Materials and Method

### 2.1. Study Design and Sample Collection

All procedures related to animal care, handling, and sampling were conducted in compliance with current legislation on animal welfare (D.Lgs. 267/2003 of the Italian Parliament). Fifty ISA Warren laying hens with an age of 20 weeks were used in this study. Hens were randomly divided into two groups of 25 each: a control group (CTR) and an experimental group (dried olive pomace; DOP). The diet of the DOP group was supplemented with dried olive pomace, whose chemical composition is reported in Table 1.

The study was conducted for a period of 28 days (after 14 days of adaptation after restocking). In this period, birds from each experimental group were housed in two separate areas in which the temperature was kept constant at 20 ± 1.5 °C, water was provided ad libitum through a common water-trough, and lighting was provided by LED bulbs of 4000 °K and 806 lumens in order to guarantee a photoperiod of 16 h of light and 8 h of darkness. All birds received an isoproteic and isoenergetic diet for laying hens in production; the diets only differed due to the 10% DOP supplementation (Table 2).

At the beginning (T0) and at the end of the experimental period (T28), the laid eggs were collected (30 for each experimental group) and immediately analyzed. Left samples were stored at −20 °C until further use.

### 2.2. Whole Blood Samples Collection

Following the 28 days of the trial, the laying hens, by now at the end of the production cycle, were transported to an authorized slaughterhouse and slaughtered in compliance with current legislation in terms of animal welfare. In this phase whole blood samples were taken from five random birds for each group. Blood samples were collected in PAXgene^®^ RNA tubes and stored at −80 °C until RNA isolation and analysis.

### 2.3. Library Preparation and RNA-Seq Analysis

Next-generation sequencing experiments, comprising RNA extraction and bioinformatics analysis, were performed by an external company (Genomix4life SRL, Baronissi, Salerno, Italy). Total RNA was extracted using TRIzol and following the manufacturer’s instructions (Invitrogen, Carlsbad, CA, USA). Indexed libraries were prepared from 300 ng/each purified RNA with a TruSeq Stranded mRNA Sample Prep Kit (Illumina, San Diego, CA, USA) according to the manufacturer’s instructions. Libraries were quantified using the TapeStation 4200 (Agilent Technologies, Santa Clara, CA, USA) and pooled such that each index-tagged sample was present in equimolar amounts, with a final concentration of the pooled samples of 2 nM. The pooled samples were subject to cluster generation and sequencing using an Illumina NextSeq 500 System (Illumina) in a 2 × 75 paired-end format at a final concentration of 1.8 pmol.

The raw sequence files generated (fastq files) underwent quality control analysis using the FastQC tool (http://www.bioinformatics.babraham.ac.uk/projects/fastqc/). To remove the adapter sequences, cutadapt was used [13]. The mapping of paired-end reads was performed using STAR (version 2.5) [14] on reference genome Oar_v3.1 (GCA_000298735.1) from Ensembl (http://www.ensembl.org/Ovis_aries/Info/Index). The quantification of transcripts expressed for each replicate of the sequenced samples was performed using the HTSeq-Count algorithm. DESeq2 [15] was used to perform the differential expression analysis. Genes with a false discovery rate (FDR) less than 0.05 were considered to be differentially expressed genes (DEGs). Raw data associated with this project are deposited in the GenBank’s Sequence Read Archive (SRA) under the Bioproject accession number PRJNA528905.

### 2.4. Enriched Pathway Analysis

Panther software (http://pantherdb.org/) was used to identify enriched pathways using the dataset of 216 downregulated genes identified between the CTR group and the DOP group with a false discovery rate (FDR) <0.05 and a LogFC <−1.5. Significance of the canonical pathway was measured with the FDR and the enrichment fold, which is generated as the ratio of the number of DEGs to the number of genes in the pathway.

### 2.5. Cholesterol Determination in Egg Yolks

The cholesterol evaluation in egg yolks was performed following the procedure recently reported by Innosa et al. [16]. The analytical method for the identification of cholesterol was calibrated in the range from 100 to 700 µg/mL (R^2^ = 0.9877).

### 2.6. Statistics

GraphPad Prism (GraphPad Software, La Jolla, CA, USA) was used for statistical analysis. Difference in egg cholesterol content between the CTR and DOP groups was assessed using Student’s *t*-test and statistical significance was considered at *p* < 0.05.

## 3. Results

### 3.1. Influence of Olive Pomace-Supplemented Diet on Whole Blood Transcriptome

In order to study the effect of dried olive pomace on the transcriptome in laying hens, we collected peripheral blood from both the control group (CTR) and the experimental group (DOP) at the end of the supplementation period. Applying a false discovery rate (FDR) <0.05, we identified 2148 differential expressed genes (DEGs) (Appendix A). This presented the possibility to clearly separate the two groups using principal component analysis (PCA)—with the first two components amounting to about 87% (Figure 1)—further confirming the homogeneity of the samples.

Moreover, on a hierarchical clustering plot, using the normalized DEG values, we were able to distinguish CTR from DOP samples (Figure 2).

To identify the biological pathways associated with DOP supplementation, we further increased stringency, applying a filter to the expression level selecting only genes with Log_2_FC either less than −1.5 or greater than +1.5. The analysis showed that 261 genes were downregulated and three genes were upregulated; because three genes was too little for an enrichment analysis, we decided to use only the downregulated genes for further analysis. All enriched pathways obtained using the downregulated genes are shown in Table 3.

Cholesterol biosynthesis was the most enriched pathway (FCE: 55.57, FDR: 2.3 × 10^−7^), followed by several pathways involved in the immune response (interferon-gamma signaling pathway, interleukin signaling pathway, and JAK/STAT signaling pathway). These results indicate that DOP supplementation can potentially reduce cholesterol and dampen inflammation. 

### 3.2. Olive Pomace-Supplemented Diet Reduced Cholesterol Egg Content

Because we found several downregulated genes belonging to the cholesterol biosynthesis pathway, we decided to assess the cholesterol content in egg yolk from the CTR and DOP groups at the end of the supplementation period. As expected, we found a significant decrease egg yolk cholesterol amount between the control group and the birds fed a diet with 10% dried olive pomace supplementation (13.91 ± 0.36 vs. 12.95 ± 0.24, *p*: 0.043) (Figure 3).

## 4. Discussion

Molecular data on the effects of DOP supplementation in chicken are still scarce. Therefore, using an RNA-seq approach, in this study we elucidated the whole blood transcriptome following a short time (28 days) of DOP supplementation in the diet of laying hens. OP belongs to the category of agro-industrial by-products which in recent years have received great attention for animal nutrition because they are rich in bioactive compounds [17,18]. In particular, OP is rich in phenolic compounds such as hydroxytyrosol, tyrosol, p-coumaric, and, to a lesser extent, vanillic acid [19,20] Olive by-products, such as leaves and pulp, have been previously used as supplementation in laying hens to evaluate their productive traits [21] and, in agreement with past studies, we did not find any difference in intake or feed conversion ratio when DOP was supplemented to the diet (data not shown). In our study, we also noted that birds from both groups remained healthy throughout the experimental period.

Peripheral blood represents an easily accessible source that has already been employed by our group [22,23,24] and others [25,26,27] to evaluate the effects of either by-products or micronutrient diet supplementation. Moreover, gene expression also reflects what occurs in non-blood tissue, making the data interpretation more flexible. Initially, we found that OP supplementation significantly regulates the expression of 2148 genes (FDR < 0.05; Appendix A), as is expected when healthy animals are fed a nutraceutical component. Thus, to ameliorate the comprehensive biological insight of our study, we performed an enrichment analysis only with genes that showed a fold change of either >1.5 or <−1.5. Using this strategy, the numbers of DEGs were reduced to 216 and three for the down- and upregulated genes, respectively. The analysis of the enriched pathways using the downregulated genes highlighted the cholesterol biosynthesis pathway as the most enriched pathway. In particular, we found the downregulation of five different genes out of the 13 involved: *Farnesyl-Diphosphate Farnesyltransferase 1* (*FDFT1*), *Farnesyl Diphosphate Synthase* (*FDPS*), *Mevalonate Diphosphate Decarboxylase* (*MVD*), *3-Hydroxy-3-Methylglutaryl-CoA Reductase* (*HMGCR*), and *3-Hydroxy-3-Methylglutaryl-CoA Synthase 1* (*HMGCS1*). HMGCS1 is the enzyme that condenses acetyl-CoA with acetoacetyl-CoA to form HMG-CoA, which is the substrate for HMGCR, the rate-limiting enzyme for cholesterol synthesis. One of the inhibitors of HMGCR is squalene [28], the content of which has been demonstrated to be elevated in olives [27]. This demonstrates that OP has a dual interference on cholesterol biosynthesis via polyphenols and squalene. FDFT1 is also considered a key enzyme of cholesterol metabolism [29] because it directs farnesyl pyrophosphate to either the sterol or non-sterol branch and is considered a potential target for new molecules for the reduction of cholesterol levels [30]. Moreover, we found that FDPS and MVD were downregulated in our dataset, probably as a consequence of the lower expression levels of HMGCS1 and HMGCR. As further confirmation of the effect of OP on cholesterol homeostasis, we found that the Regulatory Element-Binding Transcription Factor 2 (SREBF2), a transcription factor involved in the biosynthesis of lipids and cholesterol [31], was downregulated in our data (FC: −1.97, FDR: 8.93 × 10^−5^). Thus, it was not surprising that when we measured the cholesterol content in the egg yolk of both the CRTL and OP groups at the end of the supplementation time, we found a significant decrease in the experimental group (Figure 3). This finding is in agreement with a previous study where olive leaf powder supplementation showed a tendency to decrease cholesterol in the eggs of laying hens [12].

The beneficial value of olive polyphenols has been already investigated in the last decades and, as a consequence, in our study we found that DOP supplementation has an influence on several inflammatory response-related genes [32]. Thus, it was not unexpected when enriched pathways like the interferon-gamma signaling pathway, interleukin signaling pathway, JAK/STAT signaling pathway, and oxidative response were found to be downregulated (Table 3). 

## 5. Conclusions

In the present study, we clearly showed that dietary supplementation with dried olive pomace at a ratio of 10% in laying hens led to a significant decrease of cholesterol in egg yolk, probably due to the modulatory effects of polyphenolic compounds on several genes belonging to the cholesterol biosynthesis pathways. Moreover, DOP is also capable of dampening inflammation, suggesting that olive pomace diet supplementation could ameliorate chicken welfare and health.

## Figures and Tables

**Figure 1 animals-09-00427-f001:**
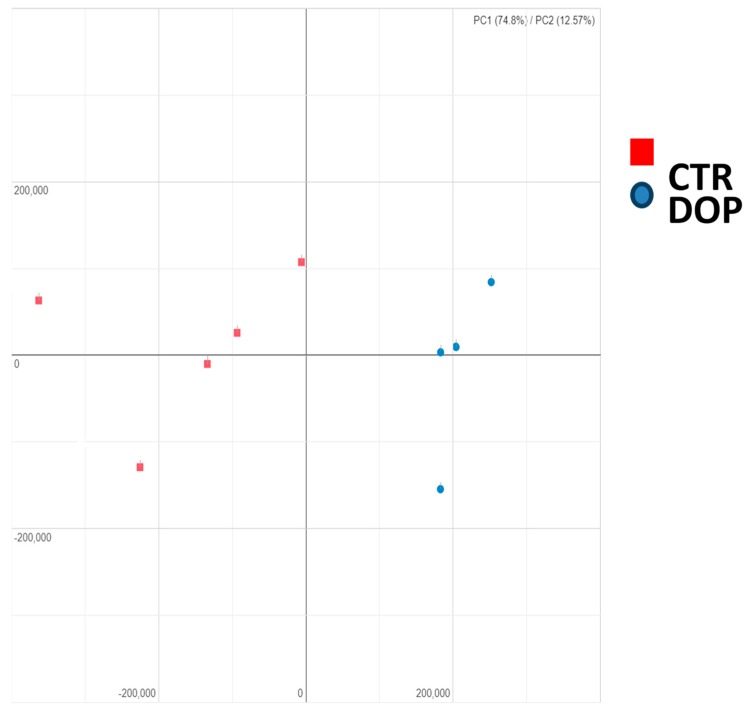
Principal component analysis (PCA) of the differentially expressed genes (DEGs) between the control group and the group fed a diet with dried olive pomace supplementation. PCA plot of both olive pomace (red squares) and control (blue circles) birds at the end of the supplementation time (28 days).

**Figure 2 animals-09-00427-f002:**
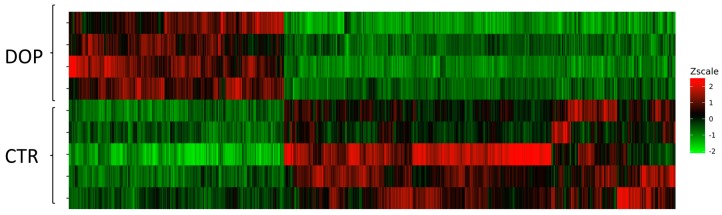
Heatmap of the differentially expressed genes (DEGs) between the control group and the group fed a diet with dried olive pomace supplementation. Changes in expression levels are displayed from green (less expressed) to red (more expressed). The order of the genes was established after hierarchical clustering using the Euclidean distance.

**Figure 3 animals-09-00427-f003:**
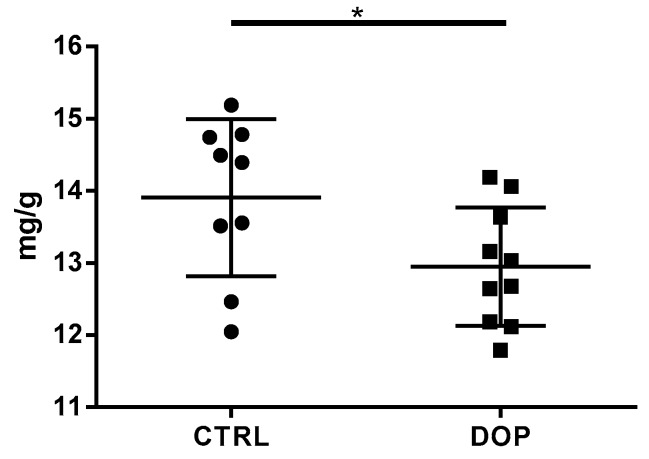
Effect of DOP-supplemented diet on egg yolk cholesterol. Cholesterol amount was recorded in individual eggs collected at the end of the supplementation time (28 days). Each dot represents a single egg and differences were analyzed using the Student’s *t*-test. * *p*-value < 0.05.

**Table 1 animals-09-00427-t001:** Chemical composition of dried olive pomace (DOP) used as dietary supplement for the laying hens of the experimental group.

Chemical Composition	DOP
Dry matter	67.24 ± 3.82
Crude protein ^1^	7.81 ± 0.66
Ether extract ^1^	15.53 ± 1.08
Neutral detergent fiber	58.13 ± 3.11

^1^ Data are reported in percentage on a dry matter basis.

**Table 2 animals-09-00427-t002:** Ingredients and composition of rations administered to laying hens fed a control diet (CTR) and a control diet supplemented with dried olive pomace (DOP).

Item	Diet	
	CTR	DOP
Ingredients (%)		
Feed for laying hens in production	78.01	83.46
Alfalfa pellet	17.56	0
Soybean meal	2.11	5.51
Granular calcium	1.16	1.12
Soybean oil	1.16	0
DOP	0	9.91
Chemical composition		
Crude protein (%, DM)	15.25	15.33
Ether extract (%, DM)	4.28	4.23
Raw cellulose (%, DM)	6.42	6.36
Ash (%, DM)	12.67	12.81
Starch (%, DM)	33.78	35.31
Lysine (%, DM)	0.81	0.83
Methionine (%, DM)	0.32	0.29
Calcium (%, DM)	3.94	3.89
Phosphorus (%, DM)	0.49	0.52
Sodium (%, DM)	0.13	0.13
Vitamin A (U.I.)	6450	6515
Vitamin E (%, DM)	19.31	19.47

DM = dry matter; I.U. = international units.

**Table 3 animals-09-00427-t003:** Panther pathways enrichment in the dried olive pomace (DOP)-supplemented group. Cholesterol biosynthesis was the most enriched Panther pathway following DOP-supplemented diet in laying hens. The analysis was conducted using the genes with a LogFc <−1.5 and a false discovery rate (FDR) <0.05. Pathways are listed based on the fold enrichment score.

Pathways	Fold Enrichment	FDR	Genes
Cholesterol biosynthesis	55.51	2.30 × 10^−7^	*FDFT1, FDPS, MVD, HMGCR, HMGCS1*
Interferon-gamma signaling pathway	29.15	7.34 × 10^−8^	*PTPN11, CISH, SOCS3, MAPK1, STAT1, JAK1 STAT2, CDKN1B, PDPK1, IL15, MKNK2*
Interleukin signaling pathway	20.29	9.46 × 10^−12^	*IL10RA, CXCR1, PIK3CA, MAPK1, STAT1, SRF, SHC1, MAPK6*
JAK/STAT signaling pathway	20.04	6.64 × 10^−3^	*STAT2, STAT1, JAK1*
Ras pathway	16.70	5.70 × 10^−8^	*ATF2, PDPK1, RHOC, PIK3CA, MAPK1, STAT1, SHC1, SRF*
Oxidative stress response	14.77	2.20 × 10^−5^	*DDIT3, ATF2, MKNK2, STAT1, MAX*
Insulin/IGF pathway/protein kinase B signaling cascade	10.93	7.06 × 10^−3^	*PDPK1, PIK3CA*
PDGF signaling pathway	10.56	6.72 × 10^−8^	*STAT1, STAT2, PDPK1, MKNK2, PIK3CA, MAPK1, SRF, JAK1, SHC1, MAPK6*
PI3 kinase pathway	10.19	2.49 × 10^−3^	*PDPK1, PIK3CA*
p53 pathway feedback loops 2	9.86	2.66 × 10^−3^	*PIK3CA, PDPK1, STAT1*
VEGF signaling pathway	8.59	4.49 × 10^−3^	*SPHK1, PIK3CA, MAPK1, MAPK6*
Apoptosis signaling pathway	8.72	7.35 × 10^−4^	*ATF2, RELA, MCM5, PIK3CA, MAPK1*
Inflammation mediated by chemokine and cytokine signaling pathway	6.81	3.13 × 10^−7^	*RELA, PDPK1, JUND, IL15, RHOC, CISH, CXCR1, PIK3CA, ARPC1B, MAPK1, STAT1, SHC1*
Angiogenesis	6.01	1.60 × 10^−4^	*PTPN11, RHOC, SPHK1, PIK3CA, MAPK1, STAT1, JAK1, SHC1, MAPK6*
CCKR signaling map	5.12	1.10 × 10^−4^	*ATF2, PDPK1, MAPK1, SRF, SHC1*
FGF signaling pathway	5.02	3.50 × 10^−2^	*PTPN11, PIK3CA, MAPK1, FGFR4*
TGF-beta signaling pathway	4.70	4.44 × 10^−2^	*ATF2, JUND, MAPK1*

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
