# Peer review of "Whole Blood Transcriptome Analysis Reveals Positive Effects of Dried Olive Pomace-Supplemented Diet on Inflammation and Cholesterol in Laying Hens"

_animals, 2019, doi:10.3390/ani9070427_

Round 1

Reviewer 1 Report

Dear authors of the manuscript “Whole blood transcriptome analysis reveals positive 2 effects of dried olive pomace supplemented diet on 3 inflammation and cholesterol in laying hens.”,

thank you for submitting your research results. Unfortunately, your manuscript need some improvements in content, presentation and, spelling as well as formatting.

ABBREVIATIONS:
- OP = olive pomace
- DOP = dried olive pomace
- DOP = experimental group fed with dried olive pomace supplementation
--> Please do not use one abbreviation for different meanings. Please be precise in the labelling within the whole manuscript!

- DOP = dried olive pomace
- OPG = ? also dried olive pomace
- DPO = ? also dried olive pomace
--> Please do not use different abbreviations for the same meaning within the manuscript!

- CTR = control group
- CG = control group
--> Please do not use different abbreviations for the same meaning within the manuscript!

-Please introduce these abbreviations within the manuscript: OPG, BHT, BSTFA, GPO

CHAPTERS AND TABLES:
Please format the numbers of chapters and tables correctly (number and boldness). There are huge mistakes all over the manuscript.

Please be consistent in spelling (e. g. down-regulated vs. downregulated)

Furthermore I have following major revisions:

Line 21: Please change the position of the words within the sentence. (several studies have been..)

Line 27:  Please introduce the abbreviation by writing (DEG) in line 26.

Line 28: Why are you using the quotation marks?

Line 40: Data of references have not to be in the text itself.

Line 42: “...oil production. Both...”

Line 45: double space before “pulp”

Line 48: “such as”

Line 55: Missing space after “percentage”

Line 55f.: Add some references for this claim.

Line 62: fatty acids are plural

Line 62: Do you mean prone or do you mean the opposite of prone? Also add a reference for this claim.

Line 79: Please support the docket number of the ethical approval commission.

Line 86: There is enough space to write “neutral detergent fibers” in the table itself. No abbreviation necessary.

Line 95: Please be consistent in using one or two decimal places within on part of the table.

Line 97: Why have you stored the eggs at -20°C and not analysed them immediately?

Line 103: Five chicken are a pretty limited number, please explain why you have not chosen more chicken.

Line 155f.: Why have you chosen the Student´s t-test?

Line 167 and 172: Please be precise! It is not “between olive pomace and control groups”, but “between control group and the group fed with dried olive pomace supplementation”. (You made such mistakes all over the manuscript.)

Line 182 f.: Please introduce first the abbreviation and not afterwards.

Line 189: Missing word

Line 203-207: This is part of the literature sector

Line 2010: If you do not show data, please write no claims here. This is not appropriate.

Line 211: missing words

Line 233: double space before “downregulated”

Line 249: It is either “laying hens” or “layers”

Line 256: Double space before “support”

Line 278 f.: Mistakes in the references!

Line 299: no double spaces necessary and in the literature source!

Line 354: You already quoted it in line 299!

You have many mistakes in formatting the references (italics, bold, points...). Please redo it precisely and consistently.

After rewriting the paper and taking more care in details, I would be pleased to review it again. But unfortunately, I have to reject the submitted manuscript.

Kind regards

Author Response

Dear authors of the manuscript “Whole blood transcriptome analysis reveals positive 2 effects of dried olive pomace supplemented diet on 3 inflammation and cholesterol in laying hens.”,

thank you for submitting your research results. Unfortunately, your manuscript need some improvements in content, presentation and, spelling as well as formatting.

Reviewer (R):

ABBREVIATIONS:

- OP = olive pomace

- DOP = dried olive pomace

- DOP = experimental group fed with dried olive pomace supplementation

--> Please do not use one abbreviation for different meanings. Please be precise in the labelling within the whole manuscript!

- DOP = dried olive pomace

- OPG = ? also dried olive pomace

- DPO = ? also dried olive pomace

--> Please do not use different abbreviations for the same meaning within the manuscript!

- CTR = control group

- CG = control group

--> Please do not use different abbreviations for the same meaning within the manuscript!

Author (A): We have revised all abbreviations consistently all through the manuscript 

R: -Please introduce these abbreviations within the manuscript: OPG, BHT, BSTFA, GPO

A: as requested by the Reviewer 2, we deleted the paragraph and left only the reference and the details about the calibration curve.

CHAPTERS AND TABLES:

R: Please format the numbers of chapters and tables correctly (number and boldness). There are huge mistakes all over the manuscript.

A: we revised the manuscript accordingly.

R: Please be consistent in spelling (e. g. down-regulated vs. downregulated)

A: we thank the reviewer for his/her comment and now we are consistent in all manuscript

Furthermore I have following major revisions:

R: Line 21: Please change the position of the words within the sentence. (several studies have been..)

A: The sentence is been revised accordingly

R: Line 27:  Please introduce the abbreviation by writing (DEG) in line 26.

A: We thank the Reviewer for his/her comment, but the abbreviation was already spelt out in the original version

R: Line 28: Why are you using the quotation marks?

A: We used the quotation mark because we used the name of the pathway as it was returned by the software. However, we have removed them because they were not changing the meaning of the sentence.

R:Line 40: Data of references have not to be in the text itself.

A: We revised the references list

R: Line 42: “...oil production. Both...”

A: the sentence is been revised

R: Line 45: double space before “pulp”

A: The double space is been revised

R: Line 48: “such as”

A: the sentence is been revised

R: Line 55: Missing space after “percentage”

A: the space is been added

R: Line 55f.: Add some references for this claim.

A: we thank the Reviewer for his/her comment and in the original version we have supported our claim with the reference Castellani et al 2017

R: Line 62: fatty acids are plural

A: We revised the sentence accordingly

R: Line 62: Do you mean prone or do you mean the opposite of prone? Also add a reference for this claim.

A: We meant that the richness of chicken meat in polyunsaturated FA makes it more prone to oxidation. In the following sentence, we used the reference of Branciari et al 2017 to support our claim

R: Line 79: Please support the docket number of the ethical approval commission.

A: As we referred to the Editor, our study does not need the ethical approvement. Our study was made in a small-sized farm located nearby an olive oil mill; during the months of October - January is the company custom to feed laying hens with a diet integrated with olive oil pomace because, based on the experience of the farm managers, eggs look better and hens show a better state of health (more uniform plumage, redder crests). Taking advantage of this management, we invited the company to divide the animals into two groups and administer the olive oil pomace only to one group. At the end of the production cycle, animals were regularly slaughtered as the company normally does in compliance with the regulation (EC) No 853/2004 of the European Parliament and of the Council of 29 April 2004 laying down specific hygiene rules for food of animal origin. In particular this regulation (Art. 1, paragraph 3d) gives the possibility to small companies of slaughtering small quantities of poultry (up to a maximum of 500 heads per year) and directly sell the products to the final consumer. The collection of whole blood samples was performed by the veterinarian responsible for monitoring the slaughtering process, avoiding any kind of alteration of the procedure. We are at your complete disposal if further clarifications are required.

R: Line 86: There is enough space to write “neutral detergent fibers” in the table itself. No abbreviation necessary.

A: the table 1 is been revised accordingly

R: Line 95: Please be consistent in using one or two decimal places within on part of the table.

A: Table 1and table 2 have been revised accordingly with the Reviewer’s requests 

R: Line 97: Why have you stored the eggs at -20°C and not analysed them immediately?

A: We apologize for any misunderstanding and we have rephrased the sentence.

R: Line 103: Five chicken are a pretty limited number, please explain why you have not chosen more chicken.

A: we Thank the Reviewer for his/her comment. We know that 5 birds are enough for our aim. Moreover, to get more stringency we have used only DEGs with Log2FC either >1.5 or <-1.5. Finally, our findings were supported by differences in egg yolk cholesterol content.

R: Line 155f.: Why have you chosen the Student´s t-test?

A: We used Student’s t-test because we measured cholesterol only at the end of supplementation time (2 groups, 1 variable).

R: Line 167 and 172: Please be precise! It is not “between olive pomace and control groups”, but “between control group and the group fed with dried olive pomace supplementation”. (You made such mistakes all over the manuscript.)

A: We thank the Reviewer for his/her comment and we revised the manuscript accordingly.

R: Line 182 f.: Please introduce first the abbreviation and not afterwards.

A: the table legend is been revised as suggested by Reviewer

R: Line 189: Missing word

A: we revised the sentence

R: Line 203-207: This is part of the literature sector

A: These lines are a small outline before the discussion and we used for making our discussion more readable and for this reason we would like to keep in this position

R: Line 2010: If you do not show data, please write no claims here. This is not appropriate.

A: We appreciate the Reviewer’s comment but we would like to keep the sentence as it is. We measured  productive and performance traits and since we do not find differences between groups we only mentioned without showing results.

R. Line 211: missing words

A: we completed the sentence

R: Line 233: double space before “downregulated”

A: We revised the double space

R: Line 249: It is either “laying hens” or “layers”

A: We thanks the Reviewer for his/her comments abut we would like to keep “laying hens”

R: Line 256: Double space before “support”

A: we deleted the double space

R: Line 278 f.: Mistakes in the references!

A: we revised the reference paragraph

R: Line 299: no double spaces necessary and in the literature source!

A: we deleted double spaces

R: Line 354: You already quoted it in line 299!

A: We apologize for the inconvenient and we deleted the reference.

You have many mistakes in formatting the references (italics, bold, points...). Please redo it precisely and consistently.

After rewriting the paper and taking more care in details, I would be pleased to review it again. But unfortunately, I have to reject the submitted manuscript.

Kind regards

Reviewer 2 Report

The rectification as mentioned are necessarily to be made for improvement of the srticle.

Review report for article no. animal-522060

Abstract:

·         Write the full form of abbreviations used at first appearance in the text (GPO, DEG, FDR etc.)

·         As depicts, the treatment group should be DOP instead of DPO (line 26, 27, 30 and in the results/discussion section –line 170)

·         Re-frame the results section in the abstract to make it more meaningful.

Introduction:

·         Give reference of olive pomace and oil yield %age from olive fruit (Line 46)

·         Add reference of Beigh et al (2015). Utilisation of apple pomace as livestock feed: A review

·         Indian Journal of Small Ruminants 21(2): 165-179

·         Add reference to the max. Level of DOP inclusion in the ration of laying chicken.

Mat. & Methods:

·         From the physical composition of diets, it is clear that the term ‘supplementation’ used in the title and in the text thereof is inappropriate, violating the basic definition in the field of Animal Nutrition. As such, the term may be replaced by more relevant term. Besides, DOP has been used as one of the ingredient in the treatment diet under present study , thus should not be confused with the term feed additive as used in Line 20-21.

·         Use P0 and P28 instead of T0 and T28 for periods.

·         Use ‘birds’ word instead of animal (Line 91 & 103)

·         No need to give the details for egg yolk cholesterol estimation procedure when the reference methodology is provide (Line 134-152 may be deleted)

Results:

·         Table 1 and Table 3 are missing from the text, although referred to.

·         Be concise in presenting the results without giving the details (delete Line 159-160)

Discussion:

·         Line-211- incomplete sentence; besides, the article would have been more informative and sensible by mentioning the productive performance results too of the study.

Author Response

Comments and Suggestions for Authors

The rectification as mentioned are necessarily to be made for improvement of the article.

Review report for article no. animal-522060

Abstract:

Reviewer (R):

Write the full form of abbreviations used at first appearance in the text (GPO, DEG, FDR etc.)

As depicts, the treatment group should be DOP instead of DPO (line 26, 27, 30 and in the results/discussion section –line 170)

Re-frame the results section in the abstract to make it more meaningful.

Authors (A):

We appreciate the Reviewer’s comments and we have revised all through the manuscript following his/her suggestions.

Introduction:

R: Give reference of olive pomace and oil yield %age from olive fruit (Line 46)

A: following the Reviewer’s suggestion, we have added reference about percentage on composition of olive fruit

R: Add reference of Beigh et al (2015). Utilisation of apple pomace as livestock feed: A review Indian Journal of Small Ruminants 21(2): 165-179

A: the suggested reference is been added in the discussion paragraph

R: Add reference to the max. Level of DOP inclusion in the ration of laying chicken.

A: We appreciate the Reviewer’s question but European Food Safety Agency has never pronounced about the maximum amount of DOP by for animal diet while there is recommendation for olive leaves(EFSA-Q-2018-00621_FAD-2010-0368_PubSum.pdf). Thus, in our study, we used a supplementation of about 10% in agreement with our previous experience with dairy cow (Castellani et al).

Mat. & Methods:

R: From the physical composition of diets, it is clear that the term ‘supplementation’ used in the title and in the text thereof is inappropriate, violating the basic definition in the field of Animal Nutrition. As such, the term may be replaced by more relevant term. Besides, DOP has been used as one of the ingredient in the treatment diet under present study , thus should not be confused with the term feed additive as used in Line 20-21.

A: We appreciate the Reviewer’s comment and for avoiding confusion, we changed the word “additive” with supplement. We would like to keep the word supplementation since we have used in our previous works when we have used either dried olive pomace (Castellani et al, 2017) or grape pomace (Iannaccone et al., 2018. However, we are open to any further suggestion and we happy to modify the article if the Reviwer thinks that the readability could be improved.

R: Use P0 and P28 instead of T0 and T28 for periods.

A: we thank the Reviewer but we would like to keep T0 and T28. We believe that those abbreviations do not confuse the readers.

R: Use ‘birds’ word instead of animal (Line 91 & 103)

A: We followed the Reviewer’s suggestion and we used birds instead of animal

R: No need to give the details for egg yolk cholesterol estimation procedure when the reference methodology is provide (Line 134-152 may be deleted)

A: According with reviewer’s suggestion, we deleted most of the methods reported in the paragraph. We left only the description of the calibration curve.

Results:

R: Table 1 and Table 3 are missing from the text, although referred to.

A: We apologize for the spelling mistake, we wrote 2 times “table 2”.

R: Be concise in presenting the results without giving the details (delete Line 159-160)

A: we appreciate the Reviewer’s suggestion but we would like to keep these 4 lines. In our opinion, it is appropriate introduce shortly how we obtained the results because it makes the paragraph more readeable.

Discussion:

R: Line-211- incomplete sentence; besides, the article would have been more informative and sensible by mentioning the productive performance results too of the study.

A: we apologize and following the Reviewer’s suggestion, we revised the sentence. We believe that show productive and performance results do not add any further information to our findings since the two groups showed no difference.

Reviewer 3 Report

The paper is very interesting and the trial is properly described. Data have been well reported as well as discussed using relevant literature. The language is also fine. In my opinion, the paper merits the acceptance. Moreover, I suggest to publish the paper in the Poultry section of Animals, where there is also an appropriate open Special Issue (i.e. Poultry Nutrition).

Author Response

Reviewer:

The paper is very interesting and the trial is properly described. Data have been well reported as well as discussed using relevant literature. The language is also fine. In my opinion, the paper merits the acceptance. Moreover, I suggest to publish the paper in the Poultry section of Animals, where there is also an appropriate open Special Issue (i.e. Poultry Nutrition).

Author:

We really appreciate the Reviewer for his/her comment. We are also proud that the Reviewer suggests to publish our data on Poultry Nutrition special issue but we decided to follow the Academic Editor suggestion’s and submit our revised manuscript to the special issue entitled: “Use of Agricultural By-Products in Animal Feeding”.
